# A Simple and Label-Free Detection of As^3+^ using 3-nitro-L-tyrosine as an As^3+^-chelating Ligand

**DOI:** 10.3390/s19132857

**Published:** 2019-06-27

**Authors:** Jin-Ho Park, Gyuho Yeom, Donggu Hong, Eun-Jung Jo, Chin-Ju Park, Min-Gon Kim

**Affiliations:** 1Department of Chemistry, Gwangju Institute of Science and Technology (GIST), 123 Cheomdangwagi-ro, Buk-Gu, Gwangju 61005, Korea; 2Center for Systems Biology, Massachusetts General Hospital, Harvard Medical School, Boston, MA 02114, USA

**Keywords:** 3-nitro-L-tyrosine, arsenic, colorimetric, label-free, chelation

## Abstract

A simple and rapid As^3+^ detection method using 3-nitro-L-tyrosine (N-Tyr) is reported. We discovered the specific property of N-Tyr, which specifically chelates As^3+^. The reaction between As^3+^ and N-Tyr induces a prompt color change to vivid yellow, concomitantly increasing the absorbance at 430 nm. The selectivity for As^3+^ is confirmed by competitive binding experiments with various metal ions (Hg^2+^, Pb^2+^, Cd^2+^, Cr^3+^, Mg^2+^, Ni^2+^, Cu^2+^, Fe^2+^, Ca^2+^, Zn^2+^, and Mn^2+^). Also, the N-Tyr binding site, binding affinity, and As^3+^/N-Tyr reaction stoichiometry are investigated. The specific reaction is utilized to design a sensor that enables the quantitative detection of As^3+^ in the 0.1–100 μM range with good linearity (R^2^ = 0.995). Furthermore, the method’s applicability for the analysis of real samples, e.g., tap and river water, is successfully confirmed, with good recoveries (94.32–109.15%) using As^3+^-spiked real water samples. We believe that our discovering and its application for As^3+^ analysis can be effectively utilized in environmental analyses such as those conducted in water management facilities, with simplicity, rapidity, and ease.

## 1. Introduction

Arsenic (As) ions are highly toxic chemical species that can cause peripheral neuropathy, skin lesions, diabetes, renal system problems, cardiovascular disease, cancer, and eventually death when exposed to the human body [1,2,3,4,5,6]. As ions occur naturally in two oxidation states of the trivalent arsenite (As^3+^) and pentavalent arsenate (As^5+^) types. These are usually considered more toxic than organic As species such as monomethylarsonic acid and dimethylarsinic acid [7,8]. Also, As^3+^ is more toxic than As^5+^ because it binds with the sulfhydryl units of proteins, which can interfere with their reactions with other enzymes and proteins [9]. This mechanism negatively influences metabolism by hindering the essential reactions between proteins in the human body or by interrupting ATP production, which can cause cell death [10]. The hazardous metal ion, As^3+^, is typically found in ground or drinking water sources because of its high aqueous solubility [11,12]. Thus, the monitoring of this harmful substance is advised and has been studied by many researchers. The World Health Organization (WHO) has established regulations that limit As^3+^ contents in water to 10 ppb [13,14].

Current techniques to determine As^3+^ levels in water include atomic absorption spectroscopy (AAS), atomic fluorescence spectroscopy (AFS), high-performance liquid chromatography (HPLC), inductively coupled plasma atomic emission spectrometry (ICP-AES), and inductively coupled plasma mass spectrometry (ICP-MS) [15,16,17,18,19]. Although these tools offer high sensitivity (below 10 ppb), they are expensive, bulky, and operationally complex (requiring operator training). Consequently, diverse approaches to overcome these drawbacks have been developed in research, including sensing strategies based on optical signals, such as absorbance or fluorescence; electrochemical signals; enzymatic reactions; and plasmonic phenomena, such as surface-enhanced Raman spectroscopy (SERS), surface plasmon resonance (SPR), or localized SPR (LSPR) [20,21,22,23,24,25].

Among the optical methods for As^3+^ detection, two main types of sensing strategies exist. The first employs particulates such as gold nanoparticles (GNP) and quantum dots (QD). Liang et al. introduced an As^3+^ sensing strategy based on the absorption of GNPs [26]. Briefly, guanine- and thymine- (G-/T)-rich single-stranded DNA (ssDNA) attached to GNP surfaces can protect the GNP from 0.1 M NaCl. However, the oligonucleotides bind As^3+^, which induces the detachment of the G-/T-rich ssDNA from the GNP. Therefore, GNP aggregation occurs in the presence of As^3+^ under 0.1 M NaCl conditions. Using this mechanism, As^3+^ concentrations could be estimated at 0.5 ppb. In addition, Zhang et al. demonstrated a turn-on As^3+^ sensing method based on the fluorescence signals of QD particles [27]. Novel fluorescent DNA QDs were synthesized by the hydrothermal treatment of G-/T-rich ssDNA. When As^3+^ reacted with these materials, a strong fluorescence signal was promoted because of the well-ordered assembly of QDs by the structural change of the oligonucleotide in the presence of As^3+^. Using this mechanism, a limit of detection (LOD) of 0.2 ppb was achieved. However, these sensing strategies require an additional reaction (Hg^2+^-EDTA) to avoid interference with other heavy metal ions or relatively long times of >12 h for the preparation of the sensing material.

The second type of sensing method involves the use of a chemical ligand. Yadav and Singh demonstrated the simple measurement of As^3+^ in water by using a synthesized “probe L” that could bind As^3+^ [28]. When this ligand reacted with the target, a color change to bright yellow was observed, and a fluorescence signal was measured after excitation at 363 nm. Based on the fluorescence intensity, the LOD for As^3+^ was reported as 66 nM. This sensing method involved a simple one-material sensing component of a ligand that chemically recognized As^3+^ by chelation. However, the selectivity of this ligand for As^3+^ was unclear because the selectivity data relating to the ligand’s reactivity with or interference by other metal ions were not shown.

Here, we discovered the specific reactivity of a natural small ligand (NSL), 3-nitro-L-tyrosine (N-Tyr), against As^3+^. Although various metal ion-chelating ligands have been developed and synthesized artificially by organic-chemical reactions [29,30,31,32,33], this chemical compound is naturally synthesized from L-Tyrosine (Tyr) in human body. Tyr is an aromatic amino acid that is moderately hydrophilic because of the hydroxyl group in its benzene substituent. As a result, Tyr is frequently exposed on protein surfaces, which allows further modifications such as nitration. Under stress, reactive oxygen species (ROS) and reactive nitrogen species (RNS) can be produced [34]. Among these, the nitrogen dioxide radical (^•^NO_2_), formed in the cascade of reactions in which the superoxide free radical (^•^O_2_^−^) reacts with nitric oxide (NO) to form peroxynitrite (ONOO^−^), which degenerates to produce ^•^NO_2_, can react with tyrosine to form N-Tyr by introducing a nitro group (NO_2_) at the *ortho* position of the phenolic ring [35,36].

In this study, we introduce a simple, rapid, and label-free sensing method for the detection of As^3+^ using N-Tyr. As shown in Figure 1, N-Tyr specifically reacts with As^3+^, producing a color change to vivid yellow. Actually, another NSL chelating Hg^2+^, which is citrate, was previously reported by our group [37]. This ligand causes the specific precipitation of the Hg^2+^-citrate complex. The precipitation must be applied to the LSPR sensing method because of its insufficient sensitivity for Hg^2+^ detection by measuring absorbance. However, unlike citrate, N-Tyr itself can be simply used as a colorimetric sensor (1 or 5 μM As^3+^ can be recognized with the naked eye). In addition, improved sensitivity (~0.1 μM As^3+^) was achieved by the absorbance measurement of As^3+^/N-Tyr mixture.

## 2. Materials and Methods

### 2.1. Materials

3-Nitro-L-tyrosine, L-tyrosine (≥98%), mercury nitrate monohydrate (≥98.5%), lead nitrate (≥99%), cadmium acetate dihydrate (98%), copper chloride (≥99%), calcium chloride (≥93%), magnesium sulfate (≥99.5%), zinc chloride (≥98%), nickel chloride hexahydrate (≥98%), manganese chloride tetrahydrate (≥98%), iron(III) chloride (97%), chromium chloride hexahydrate (96%), sodium (meta)arsenite (≥90%), and tetrabutylammonium perchlorate (TBAP, ≥99%) were purchased from Sigma-Aldrich (Milwaukee, WI, USA). Ultrapure water (UPW) with a resistivity of 18.2 MΩ·cm was obtained from a PURELAB Option-Q water-purification system (High Wycombe, Buckinghamshire, UK).

### 2.2. Absorbance Measurement

A N-Tyr solution (500 μM) and various concentrations of As^3+^ were mixed under 0.1 M acetate buffer (pH 4.0) condition. After 1 min, absorbance measurements were performed in the wavelength scan mode with 1-nm intervals using an Infinite M200 Pro/TECAN well plate reader (Männedorf, Switzerland). The absorption spectra were recorded in the range from 400 to 600 nm at 25°C. Finally, the absorption intensity at 430 nm was compared to a control solution (including N-Tyr only) for the analysis of As^3+^.

### 2.3. Infrared (IR) Measurement

A Tensor 27/Bruker Fourier-transform infrared (FT-IR) spectrometer (Billerica, MA, USA) was used to acquire IR spectra to identify the substituents participating in the binding of N-Tyr with the As^3+^. Three samples—As^3+^, N-Tyr, and a mixture of As^3+^ and N-Tyr—were individually measured in the range from 400 to 2000 cm^−1^. The measurements were performed with 1 cm^−1^ interval and 32 accumulations under air atmosphere.

### 2.4. Isothermal Titration Calorimetry (ITC) Measurement

Isothermal titration calorimetry (ITC) measurements were performed with a Nano ITC instrument (New Castle, DE, USA). Aqueous N-Tyr and As^3+^ solutions were prepared at concentrations of 0.2 and 2 mM, respectively. The samples were extensively degassed prior to titration to prevent air bubble formation during the procedure. The N-Tyr solution was placed in the sample cell of the calorimeter, and the As^3+^ was loaded in the syringe. Then, the As^3+^ solution (4.86 μL per injection) was titrated with 200-s intervals into the N-Tyr solution over 6,360 s at 25°C. Through the ITC measurement, the thermodynamic information including heating rate curve depending on time, entropy change (△S, J∙mol^−1^K^−1^), enthalpy change (△H, kJ∙mol^−1^), association constant (K_a_, M^−1^), dissociation constant (K_d_, M), and molar ratio of the reaction (*n*) was obtained.

### 2.5. Cyclic Voltammetry (CV) Measurement

Solutions of N-Tyr and As^3+^ in 90% DMF containing 0.05 M TBAP were prepared. The cyclic voltammograms of these solutions were individually measured on an indium tin oxide electrode in the potential scan range from 1.9 to −2.1 V with a scan rate of 50 mV∙s^−1^ at room temperature.

### 2.6. Spiking test

To determine the method’s applicability for real sample analysis, two real samples (tap and Yeongsan river water) were obtained locally. In these tests, the final concentrations of As^3+^ in the real samples were separately fixed as 0.1 and 10 μM. In detail, the real water samples were spiked with As^3+^ at concentrations of 0.2 and 20 μM, respectively. Then, the samples were mixed with 1 mM N-Tyr in a 1:1 volumetric ratio. After 1 min, absorption spectra were measured to determine the As^3+^ in the real samples.

## 3. Results

### 3.1. Selective Reaction Between N-Tyr and As^3+^

A ligand usually binds a metal ion through a coordinate covalent bond in which a lone pair of electrons on the ligand is donated into an empty metal orbital with polydentate bond formation (i.e., multiple bonds (2–6) are formed) and a centrally located metal ion [38]. In this work, potentially interfering metal ions such as Hg^2+^, Pb^2+^, Cd^2+^, Cr^3+^, Mg^2+^, Ni^2+^, Cu^2+^, Fe^2+^, Ca^2+^, Zn^2+^, Mn^2+^, and As^3+^ were mixed with a N-Tyr which is NSL under two conditions, either alone or as a mixture of the metal ion and As^3+^ (Figure 2a). Among these mixtures, solutions containing As^3+^ specifically turn vivid yellow, although As^3+^ is mixed with the other metal ions. However, no other metal ions produce color changes as well as blank condition (N-Tyr only). In the absorption spectra of each mixture, strong absorbances at 430 nm are observed in the presence of As^3+^ (Figure 2b). Although Cd^2+^ shows a weak absorbance increase, the rise is negligible compared to that from As^3+^. In addition, signal-to-noise (S/N) ratios, which is the relationship between metal ion/N-Tyr mixture and N-Tyr solution, are displayed. The largest intensity of S/N could be obtained in the presence of As^3+^. Based on the results, the specific reaction between N-Tyr and As^3+^ was obviously confirmed and enabled to develop the label-free sensor for As^3+^ detection without any interferences to other metal ions. The corresponding absorption spectra are shown in Appendix A.

In addition, cyclic voltammetry (CV) studies were also performed to crosscheck the binding event between N-Tyr and As^3+^ (Figure 3a). A solution of N-Tyr and As^3+^ dissolved in 90% DMF containing 0.05 M TBAP was employed in the measurement. When As^3+^ was added to N-Tyr, its CV curve was altered. Specifically, the cyclic voltammogram of N-Tyr alone exhibits a reduction peak at −1.26 V, which changes slightly to −1.28 V in the presence of As^3+^. However, during oxidation, a peak shift from 1.26 to 1.41 V occurs upon the addition of As^3+^. The new oxidation peak at 1.41 V indicates a certain interaction between N-Tyr and As^3+^.

### 3.2. Investigation of Binding Site, Affinity, and Stoichiometry

To confirm the As^3+^-binding site in N-Tyr, we first assessed the reactivity of Tyr against As^3+^, which lacks the NO_2_ group in its chemical structure—N-Tyr is formed by introducing a NO_2_ group at the *ortho* position of the phenolic ring of Tyr (Appendix A). We separately mixed various metal ions with Tyr. No changes in color or absorbance occur when Tyr is mixed with the heavy metal ions, in contrast to the vivid yellow color change observed when N-Tyr is treated with As^3+^ (Appendix A). Based on these tests, it is likely that the NO_2_ group in N-Tyr plays some role in As^3+^ chelation.

To more closely investigate the binding site, IR spectra were acquired. Three samples, i.e., As^3+^, N-Tyr, and a mixture of the two, were analyzed by their IR spectra. Prior to spectral acquisition, we predicted that the NO_2_ group of N-Tyr would participate in chelating the As^3+^ via coordination. Indeed, it is revealed that the NO_2_ group of N-Tyr acts in the chelation of As^3+^ based on a comparison of the IR peaks at 1344, 1525, and 1550 cm^−1^ (marked with asterisks, Figure 3b). The IR absorption frequency for an aromatic NO_2_ group commonly appears as two peaks in the ranges 1490–1550 cm^−1^ (N–O asymmetric stretching) and 1315–1355 cm^−1^ (N–O symmetric stretching). When N-Tyr is mixed with As^3+^, the intensities of these representative peaks decrease compared to the pristine N-Tyr. This reflects that coordination between the N-Tyr and As^3+^ restricts or interrupts the stretching motions of the NO_2_ group in N-Tyr because NO_2_ participates in the As^3+^ capturing.

Next, in order to obtain better understanding about the reaction, an isothermal titration calorimetry (ITC) measurement was conducted. An ITC measurement is commonly used to obtain information about the stoichiometry and binding affinity of a reaction between two molecules by recording a thermodynamic profile of the molecular interaction [39]. In this study, As^3+^ aliquots were sequentially injected into a solution of N-Tyr. The heat rate is dramatically increased in the initial stage of the reaction (this means that an aggressive reaction occurs between two materials), then gradually decreased through 3,200 seconds, and stabilized thereafter (Figure 3c).

The change in enthalpy (△H, kJ/mol) reflects the amount of heat released per mole of ligand bound. Thus, the quantification of the binding affinity and the determination of the reaction stoichiometry are possible. A sigmoidal curve of the △H, which typically indicates the strong binding of the molecules, is also revealed by the measurements (Figure 3d). In this test, a dissociation constant (*K*_d_) value of 4.70 × 10^−6^ M is observed. Considering that the lowest *K*_d_ value of antibody is micromolar in scale (~10^−6^ M), the binding affinity between N-Tyr and As^3+^ seems strong [40]. In addition, the *n* value, which marks the position of the inflection point on the *x*-axis in the graph of △H versus molar ratio, commonly reflects the molar ratio of the two reactants. The recorded *n* value, 0.640, indicates that two N-Tyr bind with one As^3+^ (2:1 molar ratio in the reaction). The parameter values obtained from the ITC measurement are summarized in Appendix A.

### 3.3. Optimization for As^3+^ Sensor

Prior to the quantitative analysis of As^3+^, the concentration of N-Tyr should be optimized because its original yellow color could interfere with the measurement of absorbance at 430 nm through wavelength overlap (aromatic NO_2_ compounds are commonly yellowish). Therefore, to obtain the best sensing performance that can cover the concentration regulated by WHO, N-Tyr solutions at different concentrations (50, 100, 500, and 1000 μM) were separately treated with changing concentrations of As^3+^, and then, the reacted solution were compared. In Figure 4a–d, the absorption intensity is plotted against the concentration of As^3+^. In the case of relatively lower N-Tyr concentrations such as 50 μM, signal fluctuations were showed. Unlike this, it seemed that linear responses for As^3+^ were obtained with over 100 μM N-Tyr condition. However, different sensitivities were recorded. Except for the 500 μM N-Tyr solution, zero signal was higher than that of 0.1 μM As^3+^. Only the 500 μM N-Tyr condition can distinguish 0.1 μM As^3+^ due to a relatively lower zero signal, and this distinction satisfies the As^3+^ regulation established by WHO (10 ppb ≒ 0.133 μM). Therefore, 500 μM N-Tyr was chosen for quantitative analysis of As^3+^. The corresponding absorption spectra to optimization test are displayed in Appendix A.

Further, we investigated proper pH levels for As^3+^ detection using N-Tyr (Appendix A). The stability of the developed sensing system is checked with pH variations from 3 to 11. The two conditions, N-Tyr alone and As^3+^/N-Tyr mixture, are individually exposed to different pHs. Starting weak yellow under pH 4.5, a vivid yellow color appeared in the pH range from 4.5 to 11 with N-Tyr only. In the case of the As^3+^/N-Tyr mixture, an obvious color change to yellow was shown in the pH range from 4.0 to 11. As a result, the largest absorbance difference between the two conditions, N-Tyr and As^3+^/N-Tyr mixture, was obtained under pH 4.0. Therefore, the pH 4.0 condition that can offer a clear on/off signal was selected in this study.

### 3.4. Quantitative Analysis of As^3+^

Like the test mentioned above, various concentrations of As^3+^ were mixed with the optimized N-Tyr solution. After 1 min, their respective absorption spectra were acquired. The absorbances were increased along with the concentration of As^3+^ increase (Figure 5a). Thus, As^3+^ could be quantitatively detected in a broad dynamic range, from 0.1 to 100 μM, by the differences in the absorbance after background (N-Tyr only) subtraction as shown in Figure 5b. Each As^3+^/N-Tyr mixture containing 0.1, 0.5, 1, 5, 10, 25, 50, 75, and 100 μM As^3+^ was analyzed with the calculated absorbance degrees, i.e., 0.0382, 0.0403, 0.0426, 0.0573, 0.0862, 0.1451, 0.2316, 0.3107, and 0.3844, respectively. Although a wide range was tested, good linearity (R^2^ = 0.995) was obtained and LOD (3.3σ) value reached 31.8 nM. Considering that the regulatory limit for the As^3+^ content in water is about 133 nM, the developed sensing method would be suitable for the analysis of As^3+^. With the naked eye, 1 or 5 μM As^3+^ could be determined.

Also, the stability of the reaction between As^3+^ and N-Tyr was checked (Appendix A). Three different concentrations of As^3+^ such as 0.1, 1, and 10 μM were individually reacted to N-Tyr and then stayed for 28 days under ambient condition. During the stability test, very similar absorbances could be obtained from each measurement over time.

### 3.5. Application to Analysis of As^3+^ in Real Samples

The developed sensor was applied to the analysis of As^3+^ in real water samples, such as tap and Yeongsan river water. Known concentrations of As^3+^ were artificially spiked into the real samples. The As^3+^-contaminated samples were treated to N-Tyr, and their absorption signals were subsequently measured (Table 1). In this test, the absorbances at 430 nm for the tap and river water samples were respectively 0.0365 and 0.0397 at 0.1 μM As^3+^, 0.0465 and 0.0442 at 1 μM, and 0.0831 and 0.0813 at 10 μM. These degrees were matched well to the quantitative analysis results described above with good recoveries (94.32–109.15%).

## 4. Conclusions

In summary, we discovered a kind of NSL, N-Tyr, which selectively chelates As^3+^. Using this specific reaction, a strategy for the simple and rapid detection of As^3+^ in real water samples was achieved. The proposed As^3+^ sensing method utilized the simple mixing of As^3+^ and N-Tyr. The yellow color was intensified immediately in the presence of As^3+^, and this phenomenon enabled the determination of As^3+^ within just 1 min by measurement of the absorbance at 430 nm. Using the measurement, the quantitative analysis of As^3+^ was possible over a broad dynamic range (0.1–100 μM) with good linearity (R^2^ = 0.995) and 31.8 nM of LOD. In addition, As^3+^ could be determined in real water samples from the tap and the Yeongsan River. Indeed, concentrations of 0.1, 1, and 10 μM As^3+^ in real samples were successfully recognized with good recoveries from 94.32 to 109.15%.

## 5. Discussion

In the developed sensing system, there are some limitations. First, the yellow color of N-Tyr can be overlapped to the color of the As^3+^/N-Tyr mixture. Of course, the concentration of N-Tyr was optimized for proper sensitivity in this study. However, this natural characteristic of N-Tyr has still remained a challenge. Second, the strong acidic condition (pH 4.0) of the chelating reaction can negatively affect the analysis of complicate real samples that are strong alkali and have many substances in them, such as milks and beers. Thus, we have a future plan to increase the pH value up to the neutral condition (~pH 7.0) to improve the As^3+^ sensing system.

However, despite the limitations, successful sensing properties such as broad dynamic range, good linearity, proper LOD, and real sample analysis were obviously shown. Moreover, this is the first report of As^3+^ detection based on N-Tyr chelation. We believe that our study can offer useful insight to impact broad interdisciplinary research fields as the first report of As^3+^ determination using N-Tyr and its high performances in the simplicity, rapidity, and applicability of the analysis of real water.

## Figures and Tables

**Figure 1 sensors-19-02857-f001:**
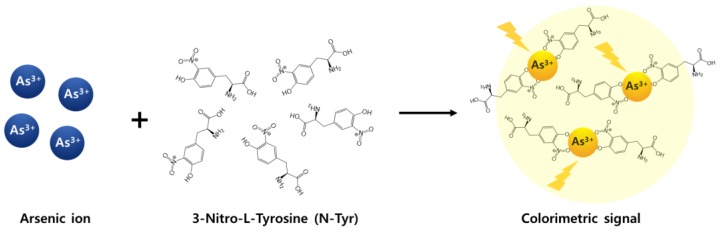
Schematic illustration of the reaction between As^3+^ and N-Tyr.

**Figure 2 sensors-19-02857-f002:**
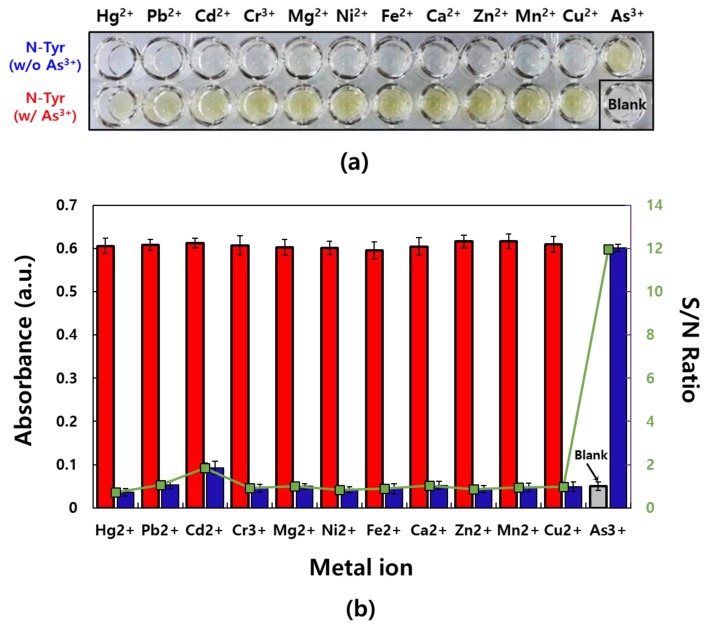
Reaction between N-Tyr and metal ions: (**a**) An image of two types of mixtures of metal ions and N-Tyr with or without As^3+^. (**b**) A graph of the absorbance values corresponding to the optical images (*n* = 6). The absorbance of mixtures of N-Tyr and metal ions with (red) or without As^3+^ (blue) are separately displayed. Also, the relationship between metal ion/N-Tyr mixture and N-Tyr solution are showed as an signal-to-noise (S/N) ratio with a secondary *y*-axis (green). The concentration of each material was 1 mM.

**Figure 3 sensors-19-02857-f003:**
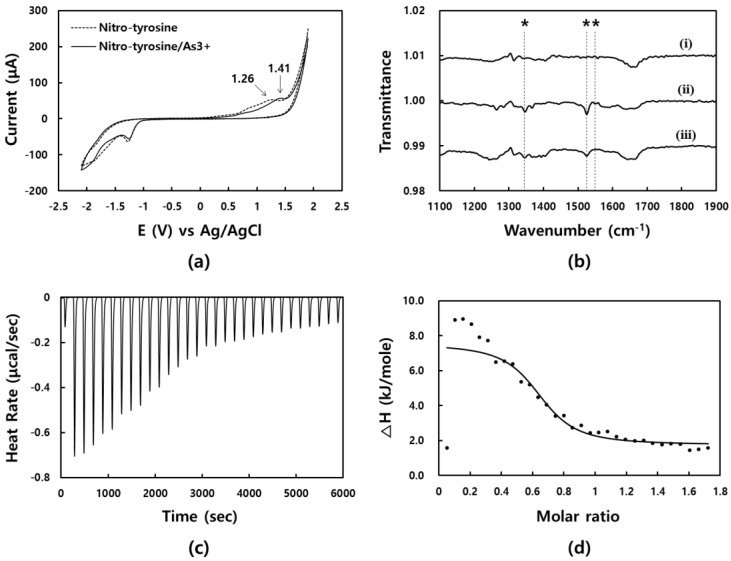
(**a**) Cyclic voltammograms of N-Tyr (dotted line) and the mixture (solid line) of N-Tyr and As^3+^: The oxidation peak is shifted from 1.26 to 1.41 V in the presence of As^3+^. (**b**) IR spectra of (i) As^3+^, (ii) N-Tyr, and (iii) As^3+^/N-Tyr. The asterisks indicate the stretching modes associated with the NO_2_ group of N-Tyr. (**c**) Isothermal titration calorimetry (ITC) graph. Heat rate along with reaction time is plotted. (**d**) The normalized fit curve of △H depending on molar ratio of the As^3+^/N-Tyr mixture is displayed.

**Figure 4 sensors-19-02857-f004:**
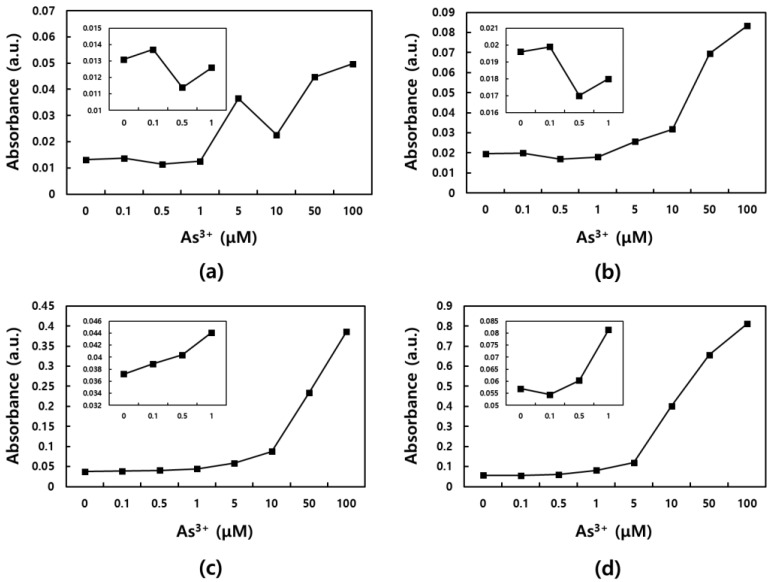
Optimization of N-Tyr concentration: Absorption spectra of As^3+^/N-Tyr mixture with (**a**) 50, (**b**) 100, (**c**) 500, and (**d**) 1,000 μM of N-Tyr. The variation of As^3+^ concentration is given from 0 to 100 μM in each test.

**Figure 5 sensors-19-02857-f005:**
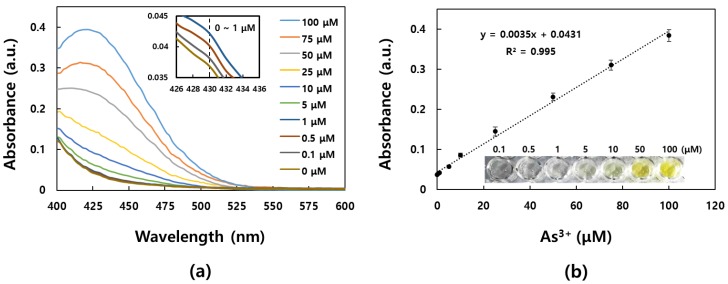
Quantitative analysis of As^3+^: (**a**) Absorption spectra as a function of the increasing concentration of As^3+^ (0.1, 0.5, 1, 5, 10, 50, and 100 μM). Inset: image of mixtures of As^3+^ and N-Tyr solution with increasing As^3+^. (**b**) Plot of the calculated absorption intensity at 430 nm corresponding to the absorption spectra against As^3+^ concentrations (*n* = 6). Inset: a magnified view of the plot in the range from 0.1 to 1 μM, with fitting equation and R^2^ value (0.995).

**Table 1 sensors-19-02857-t001:** Real sample test results obtained with the developed As^3+^ sensor (*n* = 6).

	Spiked (μM)	Measured (Abs.)	Found (μM)	Recovery (%)	RSD (±)
Tap water	0.1	0.0365	0.096	95.55	0.0004
	1	0.0465	1.091	109.15	0.0008
	10	0.0831	9.640	96.40	0.0015
Yeongsan River	0.1	0.0397	0.104	103.93	0.0007
	1	0.0442	1.037	103.76	0.0007
	10	0.0813	9.432	94.32	0.0024

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
