# Peer review of "A Simple and Label-Free Detection of As3+ using 3-nitro-L-tyrosine as an As3+-chelating Ligand"

_sensors, 2019, doi:10.3390/s19132857_

Round 1
Reviewer 1 Report
See the attachment.

Author Response
Thank you for your comments!
Please check the attached file.

Reviewer 2 Report
Dear Authors, Thank you for submitting your research article to Sensors. The paper introduced a simple and label-free method to detect As3+. The idea of this research is novel and interesting to readers. The background is well studied and the presentation of the method is very clear and sound. But I have a few questions and comments as below: 1. In the abstract, it says the minimum detectable content is 0.1 uM. Does that satisfy the WHO regulation of 10 ppb? 2. In the linearity test, can the author add more data points between 10uM and 100 uM? 3. In the discussion, it says the N-Tyr is also yellow color. How does the absorbance spectrum look like and can this spectrum be de-embedded from the As3+/N-Tyr mixture? In that case, is that possible to further improve the minimum detectable concentration of As3+? 4. Is the experiment repeatable? Thank you.Author Response
Thank you for your comments!
Please check the attached file.

Reviewer 3 Report
Dear Editor,
Authors reported a reliable determination of As3+ in tap and river waters using 3-nitro-L-tyrosine (N-Tyr). A key issue existed in the detection of As3+ was solved by using the reaction between As3+ and N-Tyr induces a prompt color change to vivid yellow, concomitantly increasing the absorbance at 430 nm. In addition, the results obtained by present method were in good agreement with those of recovery studies. This work is of great significance for environment and clinical application. This work is well written.
Therefore, I recommend considering the publication of this paper only after authors addressing the following minor points:
1. The Author should be change the decimals in the number R2=0.9962, with this valor I think is enough R2=0.996
2. Figure 2: The Authors should be included a relation between signal in the presence of As3+ and the blank signal without As3+ (S/N), in a new secondary axe, for more understanding in the figure.
3. Figure 2: Why in the interference of Cd2+ the measure of the blank is bigger than others signals?
4. How do think the Authors that they could resolve the problem of the yellow color of N-Tyr that can be overlapped to color of As3+/N-Tyr mixture.
5. How do the Authors can the determination of total arsenic? is it posible with this methodology? (As3+ + As5+)
Sincerely,
The reviewer
Author Response

(The authors gave the same response as above.)
